# Does a Polycistronic 2A Design Enable Functional FcRn Production for Antibody Pharmacokinetic Studies?

**DOI:** 10.3390/pharmaceutics17111463

**Published:** 2025-11-13

**Authors:** Valentina S. Nesmeyanova, Nikita D. Ushkalenko, Sergei E. Olkin, Maksim N. Kosenko, Elena A. Rukhlova, Ivan M. Susloparov, Dmitry N. Shcherbakov

**Affiliations:** State Scientific Center of Virology and Biotechnology “Vector”, Rospotrebnadzor, 630559 Koltsovo, Novosibirsk Region, Russia; ushkalenko@yahoo.com (N.D.U.); olkin@vector.nsc.ru (S.E.O.); kosenko_mn@vector.nsc.ru (M.N.K.); ruhlova_ea@vector.nsc.ru (E.A.R.); imsous@vector.nsc.ru (I.M.S.); dnshcherbakov@gmail.com (D.N.S.)

**Keywords:** neonatal Fc receptor, polycistronic expression, 2A peptides, pH-dependent binding

## Abstract

**Background/Objectives:** The neonatal Fc receptor (FcRn) is a heterodimeric protein composed of a heavy α-chain with an MHC class I-like fold and β_2_-microglobulin. It plays a crucial role in maintaining the homeostasis and pharmacokinetics of immunoglobulin G (IgG) and albumin through pH-dependent recycling. The production of soluble recombinant FcRn is technically challenging due to its heterodimeric structure and the presence of a transmembrane domain. This study aimed to develop a polycistronic construct enabling the co-expression of FcRn subunits from a single transcript and to evaluate the functional activity of the resulting protein in CHO-K1 cells. **Methods:** Integration vectors (pComV-FcRn-B2M) were designed to encode FcRn and β_2_-microglobulin linked via self-cleaving 2A peptides (P2A, E2A, F2A, T2A). Stable producer cell lines were generated using the Sleeping Beauty transposon system. The purified proteins were characterized by SDS-PAGE, Western blotting, and size-exclusion chromatography (SEC). Functional activity was assessed by ELISA and bio-layer interferometry (BLI). **Results:** Electrophoretic and chromatographic analyses confirmed the expected subunit composition and demonstrated that over 95% of the recombinant protein was monomeric. Functional assays revealed pH-dependent IgG binding, with strong interaction at pH 6.0 and negligible binding at pH 7.5. BLI measurements showed high affinity consistent with native FcRn function (KD = 3.15 nM at pH 6.0). **Conclusions:** The developed polycistronic construct containing a P2A peptide with a GSG linker enabled efficient production of functional FcRn in CHO-K1 cells (yield up to 2.23 mg/mL). The P2A variant demonstrated the highest efficiency and can serve as a reference system for screening Fc-engineered antibodies with optimized pharmacokinetic properties.

## 1. Introduction

The neonatal Fc receptor (FcRn), a member of the major histocompatibility complex (MHC) class I family, is essential for maintaining IgG and albumin homeostasis. By protecting these molecules from lysosomal degradation, FcRn prolongs their half-life in circulation and thereby sustains humoral immune functions [1,2,3,4]. Beyond this, it facilitates the transport of immune complexes and contributes to antigen presentation, highlighting its central role in immune regulation [5]. Owing to these functions, FcRn has become a major target for therapeutic modulation of antibody pharmacokinetics. Through Fc engineering, antibodies with either extended or reduced half-lives have been developed, enabling applications in the treatment of atopic dermatitis, Alzheimer’s disease, cancer, and infectious diseases [2,3,6,7]. FcRn is also exploited in therapies directed against autoantibody- and alloantibody-mediated disorders [8]. Moreover, Fc modifications can enhance antibody transcytosis into mucosal tissues, thereby increasing local concentrations and strengthening prophylactic protection against infection [6].

FcRn is a heterodimeric protein comprising a heavy α-chain and a soluble β2-microglobulin (β2m) subunit. The heavy chain consists of three extracellular domains (α1, α2, and α3), a transmembrane segment, and a cytoplasmic tail [1,9]. The α1–α2 domain pair forms a platform of eight antiparallel β-strands assembled into a single β-sheet, surmounted by two antiparallel α-helices [10]. Both the α3 domain and β2m exhibit an immunoglobulin-like fold. The β2m subunit is located beneath the α1–α2 platform and also interacts with the lateral surface of the α3 domain, thereby stabilizing the overall complex [6]. In human FcRn, a single N-glycosylation site has been identified within the α2 domain (Asn102), which is essential for proper expression and assembly of the molecule [11,12,13,14]. The α1–α3/β2m complex of FcRn is stabilized by an extensive network of hydrogen bonds and hydrophobic contacts, with β2m playing a key role as a stabilizing cofactor. The α1–α2 and α3 domains form the platform that supports the specific, pH-dependent interaction of FcRn with the Fc fragment of IgG [15]. Recombinant FcRn has been produced in various expression systems, including bacterial cells [16,17], mammalian cells [18,19,20], yeast [21], and insect cells [22]. Although expression in bacterial and yeast systems can yield functional protein, mammalian cells are generally preferred because they provide proper post-translational modifications, thereby ensuring full protein functionality [17,19,21,23]. A significant limitation in producing functional FcRn is the stoichiometric imbalance between the α-chain and β2m, which substantially reduces the yield of active protein [19].

Currently, two main strategic approaches are used for the expression of heterodimeric FcRn [23]. The first is based on co-transfection with two independent monocistronic vectors carrying the genes for the α-chain and β_2_-microglobulin under the control of separate promoters. In this case, the synthesis of each subunit occurs independently. The second approach involves the use of bicistronic constructs, in which the expression of each subunit gene is regulated by its own promoter but both genes are included within a single plasmid construct. In this setup, the synthesis of the individual subunits within the cell becomes coordinated. In this study, we investigated a polycistronic construct for coordinated expression of FcRn chains. For co-expression of the FcRn α-chain and β2m, a 2A peptide-based strategy was employed, as 2A peptides are effective for assembling multicomponent proteins, including antibodies [24,25]. The aim of the present work was to develop, characterize, and evaluate a polycistronic vector capable of producing functional FcRn in CHO-K1 cells.

## 2. Materials and Methods

### 2.1. Construction of Expression Plasmids pComV-FcRn_B2M Encoding 2A Peptide Variants

The pcDNA3.1 vector (Invitrogen, Carlsbad, CA, USA) served as the backbone for polycistronic expression constructs. Integration flanking arms compatible with the Sleeping Beauty (SB100X) transposon system were inserted adjacent to the CMV promoter and the neomycin resistance gene. The resulting plasmid was designated pComV.

The construct pComV-FcRn-P2A_LESS-B2M was obtained by inserting a synthetic DNA fragment into pComV using AsuNHI (SibEnzyme, Novosibirsk, Russia) and SalI (SibEnzyme, Novosibirsk, Russia) restriction sites. The expression cassette included a Gaussia luciferase (GL) leader sequence, the coding region of the FcRn heavy α-chain, a hexahistidine (6×His) affinity tag, a self-cleaving P2A peptide, and the β2-microglobulin (B2M) sequence. The cassette was codon-optimized for mammalian expression and synthesized commercially (Evrogen, Moscow, Russia).

Polymerase chain reaction (PCR) was performed with Q5 High-Fidelity DNA polymerase (New England Biolabs, Ipswich, MA, USA) using specific primers (Table 1). Amplification was carried out on a Veriti thermal cycler (Thermo Fisher Scientific, Waltham, MA, USA) under the following conditions: 30 cycles of 95 °C for 30 s, 62 °C for 20 s, and 72 °C for 1 min. PCR products were gel-purified (Qiagen, Hilden, Germany), ligated with T4 DNA ligase (SibEnzyme, Novosibirsk, Russia), and transformed into chemically competent E. coli NEBStable cells (New England Biolabs, Ipswich, MA, USA). Transformants were selected on LB agar containing kanamycin (50 µg/mL). Plasmid sequences were confirmed by Sanger sequencing (Genomics Core Facility, SB RAS, Novosibirsk, Russia).

To generate the constructs pComV-FcRn-P2A-B2M, pComV-FcRn-E2A-B2M, pComV-FcRn-F2A-B2M, and pComV-FcRn-T2A-B2M, two-step overlap extension PCR was used. In the first step, two fragments were amplified: (i) the FcRn fragment with the respective 2A peptide sequence at the 3′ end, and (ii) the B2M fragment containing the same 2A peptide sequence at the 5′ end, preceded by a GSG linker and a Furin cleavage site (Table 1). In the second step, the complete cassette was assembled and amplified by overlap extension PCR. The resulting products were cloned into the pComV vector using EcoRI (SibEnzyme, Novosibirsk, Russia) and SalI (SibEnzyme, Novosibirsk, Russia) restriction sites. Construct sequences were confirmed by Sanger sequencing (Genomics Core Facility, SB RAS, Novosibirsk, Russia).

### 2.2. Stable Transfection of CHO-K1 Cells with Constructed Plasmids

Stable transfection of CHO-K1 cells was performed using the “Nucleic Acid Transfection in Eukaryotic Cells with PEI” kit (Biospecifica LLC, Novosibirsk, Russia). A suspension culture of CHO-K1 cells was maintained in HyCell CHO growth medium (Cytiva, Marlborough, MA, USA) until reaching 8 × 10^6^ viable cells/mL. For each transfection, 5 × 10^6^ viable cells were collected by centrifugation and resuspended in TransFx-C transfection medium (Cytiva, Marlborough, MA, USA). Following the manufacturer’s protocol, cells were transfected with a mixture of one construct (pComV-FcRn-B2M-LESS-GSG, pComV-FcRn-P2A-B2M, pComV-FcRn-E2A-B2M, pComV-FcRn-F2A-B2M, or pComV-FcRn-T2A-B2M) and the pSB100X plasmid encoding the Sleeping Beauty transposase at a mass ratio of 10:1 (construct:pSB100X).

### 2.3. Generation of CHO-FcRn-P/L, CHO-FcRn-P2A, CHO-FcRn-E2A, CHO-FcRn-F2A, and CHO-FcRn-T2A Producer Cell Lines

To establish producer cell lines expressing different FcRn constructs, Geneticin G418 (PanEco, Moscow, Russia) was applied to the culture medium 72 h post-transfection at 250 µg/mL. Cells were maintained in fresh HyCell CHO medium (Cytiva, Marlborough, MA, USA) in a Heracell Vios 160i incubator (Thermo Fisher Scientific, Waltham, MA, USA) at 37 °C with 5% CO_2_ on an orbital shaker (Infors, Bottmingen, Switzerland; 180–200 rpm, 19 mm orbit). Stepwise selection was performed over 12 days by increasing the G418 concentration every 72 h in 50 µg/mL increments up to a final concentration of 500 µg/mL. Cell viability was monitored at each stage using trypan blue exclusion and an automated cell counter (Bio-Rad, Hercules, CA, USA). Following selection, cultures were maintained for 3 additional days under the same conditions to stabilize growth. Cells were then cryopreserved at 5 × 10^6^ cells/mL in a freezing medium composed of 50% FBS (Thermo Fisher Scientific, Waltham, MA, USA), 40% growth medium, and 10% DMSO (PanEco, Moscow, Russia).

### 2.4. Culture Conditions for CHO-FcRn-P/L, CHO-FcRn-P2A, CHO-FcRn-E2A, CHO-FcRn-F2A, and CHO-FcRn-T2A Producer Cell Lines

Frozen producer cells (CHO-FcRn-P/L, CHO-FcRn-P2A, CHO-FcRn-E2A, CHO-FcRn-F2A, CHO-FcRn-T2A) were thawed and transferred into 50 mL tubes containing 5 mL of HyCell CHO medium (Cytiva, Marlborough, MA, USA) at a starting density of 1 × 10^6^ cells/mL. Cells were cultured in a Heracell Vios 160i incubator (Thermo Fisher Scientific, Waltham, MA, USA) at 37 °C with 5% CO_2_ for 3 days. Upon reaching a density of 10–13 × 10^6^ cells/mL, cells were seeded into 250 mL flasks containing 50 mL of medium at 1 × 10^6^ cells/mL and cultured at 37 °C with 5% CO_2_ on an orbital shaker (Infors, Bottmingen, Switzerland; 180–200 rpm, 19 mm orbit). Cell density and viability were measured 48 h after seeding using trypan blue exclusion and an automated cell counter (Bio-Rad, Hercules, CA, USA). When cultures reached 6–8 × 10^6^ cells/mL, the incubation temperature was reduced to 31 °C, and cultivation continued for 9–10 days in a CO_2_ incubator (Heracell Vios 160i, Thermo Fisher Scientific, Waltham, MA, USA). At the end of the incubation period, cells were harvested by centrifugation at 10,000 rpm for 15 min at 4 °C, and supernatants (50 mL per flask) were collected for subsequent analyses.

### 2.5. Isolation and Purification of Recombinant FcRn by Affinity Chromatography

His-tagged recombinant FcRn was purified by affinity chromatography using a Ni-NTA resin (Cytiva, Marlborough, MA, USA). A pre-packed 1 mL nickel-charged column was equilibrated with 10 column volumes of phosphate-buffered saline (PBS; Servicebio, Wuhan, China; pH 7.4). For protein binding, the resin was removed from the column and incubated overnight at 4 °C with 20 mL of protein sample under gentle mixing. The resin was then reloaded into the column, and purification was continued. The column was washed with 20 column volumes of PBS containing 20 mM imidazole (CDH, New Delhi, India) to remove nonspecifically bound proteins. Target protein was eluted with PBS supplemented with 300 mM imidazole, applying 1 mL of elution buffer to the column. After elution, the column was washed with 10 column volumes of distilled water, regenerated with 0.1 M NaOH, and rinsed again with distilled water. The resin was stored in 20% ethanol without stripping nickel ions to allow reuse for purification of the same protein. Fractions obtained during washing and elution were collected and analyzed by SDS-PAGE to assess protein purity and yield.

### 2.6. Protein Concentration Assessment

The concentration of purified FcRn was determined spectrophotometrically at 280 nm using a NanoDrop instrument (Thermo Fisher Scientific, Waltham, MA, USA). The theoretical extinction coefficient (ε) was calculated with ExPASy ProtParam based on the amino acid sequence of FcRn, considering both the α-chain and β2-microglobulin subunit. Measurements were performed in three independent replicates, and the mean value, standard deviation (SD), and standard error of the mean (SEM) were calculated using GraphPad Prism (version 8, GraphPad Software, San Diego, CA, USA).

### 2.7. Size-Exclusion Chromatography (SEC)

The oligomeric state of purified FcRn proteins was analyzed by size-exclusion chromatography (SEC) under non-denaturing conditions. A 10 μL protein sample (0.5 mg/mL) was loaded onto serially connected PROTEIN KW-803 columns (8.0 mm ID × 300 mm; Shodex, Tokyo, Japan). Chromatographic separation was performed on an LC-20 Prominence system controlled by LabSolutions software (version v5.95, Shimadzu, Kyoto, Japan). The column temperature was maintained at 25 °C. The mobile phase consisted of 0.05 M NaH_2_PO_4_·2H_2_O and 0.3 M NaCl, adjusted to either pH 6.0 or pH 7.4 with NaOH. The flow rate was 0.7 mL/min, and protein elution was monitored at 214 nm. Calibration was performed using proteins of known molecular weight: β-lactoglobulin (35 kDa; PSS Polymer Standard Service GmbH, Mainz, Germany), albumin (44 kDa; PSS Polymer Standard Service GmbH, Mainz, Germany), and myoglobin (17.5 kDa; PSS Polymer Standard Service GmbH, Mainz, Germany). Apparent molecular weights of FcRn were estimated using calibration curves obtained separately under pH 6.0 and pH 7.4 conditions. The relationship between lg(molecular weight) and retention time (Ret. Time) for standard proteins was described by the following equations:at pH 6.0: lg(MW) = −0.1751 × Ret. Time + 6.697at pH 7.4: lg(MW) = −0.1777 × Ret. Time + 6.768

### 2.8. Western Blotting

Purified FcRn samples were separated by SDS-PAGE under denaturing conditions on a 12% polyacrylamide gel and transferred to a nitrocellulose membrane (Bio-Rad, Hercules, CA, USA) for 10 min at room temperature using Tris–Glycine buffer (pH 8.3) supplemented with 20% methanol. The membrane was blocked with 5% (*w*/*v*) BSA in PBST (PBS containing 0.2% Tween-20) for 15 min at room temperature. For detection, the membrane was incubated with Anti-Beta-2 Microglobulin Rabbit Polyclonal Antibody (1:130; Cloud-Clone Corp., Katy, TX, USA) for 10 min, washed with PBST (3 × 10 min), and then incubated with Polyclonal Antibody to Fc Fragment of IgG Receptor Transporter Alpha (FCGRT) (1:500; Cloud-Clone Corp, Katy, TX, USA) for 10 min, followed by washing with PBST (3 × 10 min). Subsequently, the membrane was treated with Goat anti-Rabbit IgG (H + L)-HRP secondary antibody (1:10,000; Abcam, Cambridge, UK) for 10 min. After a final wash with PBST (3 × 10 min), protein bands were visualized using Pierce 1-step Ultra TMB Blotting Solution (Thermo Fisher Scientific, Waltham, MA, USA). The reaction was stopped by rinsing the membrane with distilled water after approximately 10 min.

### 2.9. Enzyme-Linked Immunosorbent Assay (ELISA)

ELISA was performed with purified samples obtained after affinity chromatography. Monoclonal antibodies 6b3 or 900 (100 ng per well in 1× PBS, pH 7.5) were used as antigens, coated onto 96-well Nunc plates ( Thermo Fisher Scientific, Waltham, MA, USA), and incubated at 4 °C for 20 h. All subsequent steps were performed at two pH conditions: 6.0–6.1 or 7.4–7.5. After washing, plates were incubated with PBST (0.05% Tween-20, 200 µL per well), blocked with 0.2% casein in PBS for 1 h at 37 °C and 600 rpm, and then incubated with test samples diluted in the same buffer for 30 min at 37 °C and 600 rpm. Secondary antibodies conjugated with horseradish peroxidase (HRP Anti-6×His Tag Antibody, Abcam, Cambridge, UK; dilution 1:10,000) were added and incubated for 30 min at 37 °C and 600 rpm. Following the final wash, TMB substrate (TransGen Biotech Co., Ltd., Beijing, China; 100 µL per well) was added, and the reaction was stopped by addition of 1 M H_2_SO_4_ (50 µL per well). Absorbance was measured at 450 nm.

### 2.10. Bio-Layer Interferometry (BLI)

The kinetics of FcRn–IgG interactions were analyzed by bio-layer interferometry (BLI) using an Octet RED96e system (ForteBio, Fremont, CA, USA). Recombinant human FcRn variants (FcRn-P2A, FcRn-E2A, and FcRn-F2A) were used as ligands, and the single-chain antibody IgG 900 (IgG1, ~55 kDa) served as the analyte. HIS1K biosensors (Anti-Penta-HIS, Sartorius, Göttingen, Germany) were pre-equilibrated in phosphate-buffered saline (PBS, pH 7.4) and subsequently loaded with FcRn at a concentration of 5 µg/mL. Association with IgG 900 was monitored at a concentration of 89.9 nM, followed by dissociation in the same buffer.

Binding kinetics were evaluated at two pH values (pH 6.0–6.1 and pH 7.4–7.5) to characterize the pH dependence of FcRn–IgG interactions. Data acquisition and fitting were performed using ForteBio Data Analysis software (version 12.0, ForteBio, Fremont, CA, USA) with a global 1:1 binding model. Statistical analysis were performed using R (version 4.3.1, R Foundation for Statistical Computing, Vienna, Austria) in RStudio (version 2023.12.1+402, Posit Software, PBC, Boston, MA, USA).To evaluate the significance of differences in KD values between pH 6.0 and 7.5, and among the human FcRn variants (FcRn-P2A, FcRn-E2A, and FcRn-F2A), 95% confidence intervals of the mean values were calculated. Non-overlapping confidence intervals were interpreted as indicating statistically significant differences (approximately corresponding to *p* < 0.05).

## 3. Results

The production of a soluble recombinant analogue of FcRn presents several challenges. First, FcRn is a heterodimer consisting of a heavy α-chain with an MHC I-like fold and β2-microglobulin. Its expression therefore requires either two separate vectors or specialized constructs enabling the simultaneous synthesis of both protein components. Second, FcRn is a transmembrane protein, and a soluble form can only be generated by removal of the transmembrane domain. Taking these features into account, the amino acid sequence was designed (Figure 1) to yield a product capable of cleavage through the incorporation of a 2A peptide.

Differences in the primary sequence of the synthesized product were introduced by the 2A peptide, with four variants used (P2A, E2A, F2A, T2A). In most constructs, a GSG linker was included upstream of the sequence encoding the 2A peptide. One construct lacked the linker sequence and was used for comparison (Table 2).

For expression of the designed sequence, the pComV-FcRn-B2M vector was constructed. This plasmid is a modified derivative of pcDNA3.1, adapted for stable genomic integration in mammalian cells. The expression cassette contains the FcRn and B2M genes, linked by a sequence encoding one of several 2A peptide variants, under the control of the CMV promoter. Neomycin resistance (G418) was provided by a gene driven by the SV40 promoter. In addition, the vector contains flanking sequences compatible with the Sleeping Beauty transposon system (Figure 2).

The constructs were used for stable transfection of CHO-K1 suspension cells. Polyclonal producer lines (CHO-FcRn-P/L, CHO-FcRn-P2A, CHO-FcRn-E2A, CHO-FcRn-F2A, and CHO-FcRn-T2A) demonstrated stable population doubling over 6–7 passages, reaching the working cell density (10–13 × 10^6^ cells/mL) with high viability (95–98%). Cultivation was carried out for 12–13 days under standard conditions (37 °C), followed by a temperature shift to 31 °C to enhance protein expression. FcRn proteins were harvested from the culture supernatant and purified by Ni-NTA affinity chromatography. Elution was performed in PBS (pH 7.4) containing 300 mM imidazole. The purified samples (FcRn-P/L, FcRn-P2A, FcRn-E2A, FcRn-F2A, and FcRn-T2A) were analyzed by SDS–PAGE and Western blotting to assess molecular weight and subunit composition (Figure 3a,b).

Recombinant FcRn obtained from the Immunochemistry Laboratory of the Federal Research Center for Virology and Microbiology “Vector” (Rospotrebnadzor) was used as a positive control. Commercial BSA (SibEnzyme LLC, Novosibirsk, Russia) served as a negative control. Denaturing SDS–PAGE analysis indicated that the expressed protein consists of two subunits with approximate molecular weights of ~35 kDa (heavy α-chain) and ~14 kDa (β2-microglobulin). Bands at the expected positions were detected in FcRn-P2A, FcRn-E2A, and FcRn-F2A samples, whereas no specific bands were observed in FcRn-P/L and FcRn-T2A samples. Western blot analysis yielded similar results and additionally revealed a band at ~42 kDa, potentially corresponding to the uncleaved polyprotein.

The concentrations of recombinant proteins in the FcRn-P2A, FcRn-E2A, and FcRn-F2A samples were determined. Protein measurements were performed in three technical replicates for each polyclonal pool, and the results are presented in Figure 3c. The highest concentration was observed for FcRn-P2A (2.23 mg/mL), which also exhibited slightly higher variability compared with the other constructs (SD = 0.235 versus 0.072 and 0.024 for FcRn-E2A and FcRn-F2A, respectively). The standard error of the mean (SEM) is shown in the figure to indicate the precision of the mean estimate. FcRn-E2A and FcRn-F2A displayed similar expression levels (~1.7 mg/mL) with high reproducibility, reflecting their stability under the experimental conditions.

Size-exclusion chromatography (SEC) was employed to assess the oligomeric state and conformational heterogeneity of the purified recombinant FcRn variants under native conditions. Proteins of known molecular weight were used as calibration standards: myoglobin (17.5 kDa), β-lactoglobulin (35 kDa), and chicken albumin (44 kDa). Considering the pH-dependent function of FcRn–binding IgG at acidic pH (~6.0) in endosomes and dissociating at physiological pH (~7.5) in the bloodstream—analyses were performed under both conditions.

At pH 6.0, SEC profiles of FcRn-P2A, FcRn-E2A, and FcRn-F2A samples exhibited a pronounced main peak, representing 98–99% of the total area, at a retention time of 29.1–29.2 min. This peak presumably corresponds to monomeric FcRn with a calculated molecular weight of ~37.9–38.9 kDa (Figure 4).

Minor peaks accounted for no more than 1–2% of the total area. In the FcRn-P2A and FcRn-F2A samples, these peaks corresponded to small fragments (~5.7 and ~2.0 kDa), whereas in the FcRn-E2A sample a distinct peak of ~110 kDa (≈1%) was detected, most likely representing dimer or aggregate formation. Overall, no significant differences among the samples were observed at pH 6.0.

At pH 7.4, the chromatographic profiles likewise showed predominantly monomeric FcRn (95–97% of the total area; calculated MW ≈ 38.9–41.9 kDa) (Figure 5).

Minor additional peaks were classified into two categories: (1) high-molecular-weight aggregates or dimers (112–231 kDa, up to 3% of the total area), most pronounced in the FcRn-E2A variant, and (2) low-molecular-weight fragments (~2–7 kDa, 1–3%). The FcRn-P2A and FcRn-F2A samples exhibited greater stability, whereas higher levels of both types of impurities were observed in the FcRn-E2A sample.

Overall, FcRn remained predominantly monomeric at both pH 6.0 and 7.4. Observed differences between the two pH conditions were minimal and are most likely attributable to alterations in hydrodynamic properties and charge rather than significant structural rearrangements. Nevertheless, the FcRn-E2A variant demonstrated an increased tendency for aggregate and fragment formation compared with FcRn-P2A and FcRn-F2A.

At the previous stage, it was confirmed that the purified FcRn variants predominantly existed in a monomeric form under physiologically relevant pH conditions. The next step was to evaluate their functionality by assessing their ability to specifically interact with IgG.

For this purpose, ELISA was performed using two monoclonal antibodies, 6b3 and 900, as model ligands. Antibody 6b3 is a chimeric protein consisting of the nanobody Nb6 [26] fused to the Fc fragment of human IgG1 (CH2–CH3 domains containing the M252Y, S254T, T256E, E333A, and H433K/N434F mutations, which extend antibody half-life).

Antibody 900 is a chimeric protein composed of the variable regions of the light and heavy chains of the murine antibody 9E2 fused to the Fc fragment of human IgG1 (CH2–CH3 domains carrying the same set of mutations: M252Y, S254T, T256E, E333A, and H433K/N434F) [27]. It is well established that FcRn exhibits pH-dependent binding: at acidic pH, it forms a complex with IgG, whereas under neutral conditions the complex dissociates [28]. To determine whether this property was preserved in the recombinant analogs, ELISAs were conducted at pH 6.0 and pH 7.5. The analysis included a positive control (recombinant FcRn kindly provided by the Laboratory of Immunochemistry, State Research Center of Virology and Biotechnology “Vector”, Rospotrebnadzor) and a negative control (PBS). Figure 6 presents the comparative binding data of monoclonal antibodies 6b3 and 900 with the recombinant FcRn variants under the tested conditions.

All tested FcRn variants (P2A, E2A, and F2A) demonstrated a pronounced pH-dependent binding pattern with both antibodies: strong binding was observed at pH 6.0, whereas the signal dropped to background levels at pH 7.5. The mean optical density (OD) values at neutral pH were close to background (0.05–0.06), indicating minimal nonspecific interaction. For antibody 6b3, the highest signal was observed for the FcRn-P2A variant (OD 2.8), whereas FcRn-E2A and FcRn-F2A showed lower binding levels (OD 1.9–2.0). A similar trend was noted for antibody 900: the FcRn-P2A variant exhibited the strongest binding (OD ~3.1), while FcRn-F2A and FcRn-E2A showed slightly reduced signals (OD 2.5–2.6).

To obtain a more accurate assessment of the binding affinity of the three FcRn variants (P2A, E2A, and F2A) toward antibody 900 under different pH conditions, bio-layer interferometry (BLI) was performed. The main kinetic parameters of the interactions are summarized in Table 3. Representative binding sensograms of antibody 900 interacting with immobilized FcRn at different pH values are provided in Appendix A.

Binding constant measurements demonstrated that all three FcRn variants interact with antibody 900 with high affinity at pH 6.0, with KD values ranging from 3.15 to 4.13 nM. At neutral pH 7.5, binding was markedly weaker, with KD values of 14.3–17.3 nM. Statistical analysis was performed using pairwise comparisons of mean values with 95% confidence intervals (CIs). At pH 6.0, FcRn-P2A (KD = 3.15 nM, 95% CI: 2.93–3.37) and FcRn-E2A (KD = 3.46 nM, 95% CI: 3.29–3.63) exhibited significantly higher affinity compared to FcRn-F2A (KD = 4.13 nM, 95% CI: 3.87–4.39), while the difference between FcRn-P2A and FcRn-E2A was not significant. At pH 7.5, the KD values of all variants were in a similar range (14.3–17.3 nM), with no statistically significant differences observed. For evaluating differences in affinity among the FcRn variants at each pH, pairwise comparisons using 95% confidence intervals were also performed. At pH 6.0, FcRn-P2A showed a significantly lower KD value (3.15 nM, 95% CI: 2.93–3.37) compared to FcRn-F2A (4.13 nM, 95% CI: 3.87–4.39; *p* < 0.05), indicating higher binding affinity for FcRn-P2A. Similarly, the KD value for FcRn-E2A (3.46 nM, 95% CI: 3.29–3.63) was also significantly lower than that of FcRn-F2A (*p* < 0.05). In contrast, the difference between FcRn-P2A and FcRn-E2A did not reach statistical significance (*p* > 0.05). At pH 7.5, the KD values for all three variants were comparable (14.3–17.3 nM), and no statistically significant differences were detected (*p* > 0.05).

## 4. Discussion

FcRn is a key element in biomedical research focused on the targeted modulation of antibody pharmacokinetics. The use of recombinant FcRn analogs enables the assessment of therapeutic molecules’ affinity for the Fc fragment of IgG, which correlates with their in vivo half-life and is widely applied in the development of Fc-engineered antibodies. However, the production of functionally active FcRn is complicated by its heterodimeric nature and the presence of a transmembrane domain. The latter challenge is typically addressed by designing truncated, soluble forms lacking this domain. Previous strategies for producing recombinant soluble FcRn primarily relied on separate vectors for the co-expression of the α-chain and β_2_-microglobulin in mammalian cells [10,18,29]. Alternative approaches have also included the use of dual-promoter constructs [21,23]. In contrast to these strategies, the present study reports, to the best of our knowledge, the first successful production of functional FcRn using a polycistronic construct incorporating 2A peptides. This approach enabled the co-expression of the FcRn α-chain and β2-microglobulin from a single transcript, resulting in a stable and functionally active protein.

Analysis of the literature indicates that the most efficient 2A peptides for protein processing are generally considered to be P2A and T2A, although their relative efficiency depends on several factors, including the expression system, the nature of the target protein, the organization of the expression cassette, and the specific 2A sequence context (e.g., size and presence of linkers). For instance, Liu et al. systematically optimized the use of 2A peptides in polycistronic vectors and reported preferential activity of T2A in bicistronic constructs [30], whereas Kim et al. noted the advantage of P2A in various cell cultures [31]. In experiments with CHO cells expressing antibodies, Chng et al. demonstrated that T2A provides the highest cleavage efficiency [32]. Li et al. compared P2A and T2A for co-expression of GFP and RFP in CHO cells and found that the P2A-containing construct resulted in higher accumulation of target proteins compared to T2A [33]. Collectively, these studies confirm that the choice of an optimal 2A peptide requires task- and system-specific optimization [31]. Our experiments show that polycistronic constructs containing P2A, E2A, and F2A peptides enable the production of correctly assembled soluble FcRn. All constructs included a GSG linker between the Furin cleavage site and the 2A peptide sequence. Attempts to produce a construct lacking this linker did not yield detectable protein: for the P2A variant without the GSG linker, no protein was observed by either SDS-PAGE or Western blot. These observations are consistent with previous reports indicating that inclusion of a GSG linker enhances ribosomal skipping efficiency [32,34,35]. An exception was the T2A construct, which contained a GSG linker but still failed to produce detectable protein in SDS-PAGE or Western blot. It should be noted that our approach has inherent limitations. Although the Sleeping Beauty transposon system allows stable expression through integration into transcriptionally active genomic regions, the integration is partially random, limiting the predictability of expression levels [36,37].

The construct containing P2A demonstrated a higher level of recombinant protein expression compared to the other variants; however, this was accompanied by greater variability between replicates. In contrast, the E2A- and F2A-containing constructs exhibited lower expression levels but higher reproducibility. These results are based on analyses of single polyclonal pools for each construct and therefore should be considered as preliminary observations. A definitive assessment will require further investigation using independent clonal cell lines.

Size-exclusion chromatography (SEC) analysis demonstrated that FcRn variants produced using P2A, E2A, and F2A peptides predominantly exist in a monomeric form, consistent with their native organization. The chromatographic profiles obtained at pH 6.0 and 7.4 showed no substantial differences, suggesting that the proteins retain their native subunit folding and functional integrity within this pH range. For the E2A construct, a higher proportion of aggregates and low-molecular-weight fragments was detected (~2% at pH 6.0 and ~5% at pH 7.4), which may limit its applicability in cases where protein stability and homogeneity are critical. From a practical standpoint, the predominance of the monomeric form and the pH stability of the complex are key parameters for the development of recombinant and therapeutic proteins.

Functional activity analysis demonstrated that all FcRn variants (P2A, E2A, and F2A) retained the key physiological property of the receptor—pH-dependent binding to IgG. Under acidic conditions (pH 6.0), specific binding was observed, whereas at neutral pH (7.5) the signal decreased to background levels, consistent with previous reports describing the mechanism of FcRn–IgG interaction [16,17,19,21,23]. Among the tested variants, FcRn-P2A exhibited the strongest binding to both antibodies 6b3 and 900. These differences were most pronounced at lower dilutions, where FcRn-P2A signals consistently exceeded those of FcRn-E2A and FcRn-F2A. In electrophoretic and Western blot analyses, this variant also appeared more homogeneous. In contrast, the FcRn-E2A variant showed a higher proportion of aggregates and low-molecular-weight fragments, as confirmed by SEC data. It is possible that the presence of such aggregates affected the ELISA binding signal. Although the FcRn-F2A variant demonstrated a degree of homogeneity comparable to that of FcRn-P2A according to SEC, its ELISA signal intensity was lower. Bio-layer interferometry (BLI) results further revealed that the dissociation constants (KD) for all three FcRn variants at pH 6.0 were within a narrow range (3.15–4.13 nM). At neutral pH (7.5), binding affinity was markedly reduced (KD = 14.3–17.3 nM), reflecting the typical physiological behavior of FcRn, characterized by pH-dependent interaction with IgG. At pH 6.0, the FcRn-P2A and FcRn-E2A variants exhibited higher binding affinity compared to FcRn-F2A, whereas the difference between FcRn-P2A and FcRn-E2A was not significant. At pH 7.5, no differences in affinity were observed among the variants. Thus, BLI results quantitatively confirm the pH-dependent binding observed in ELISA and demonstrated that the observed differences between FcRn variants were not attributable to changes in binding affinity.

ELISA results showed that antibody 900 exhibited stronger binding to the recombinant FcRn variants compared to antibody 6b3, despite both carrying identical Fc fragment mutations that enhance FcRn affinity. These differences are likely attributable to structural characteristics of the variable domains and their influence on the conformational flexibility of IgG, which may, in turn, affect the accessibility of the Fc region for FcRn interaction [38].

However, a limitation of our study should be considered. We have demonstrated the functionality of the soluble FcRn form in vitro, but it lacks the transmembrane domain and is therefore incapable of replicating the full cellular IgG recycling cycle. This prevents direct modeling of in vivo antibody transport in our system. It is important to emphasize that the polycistronic vector we developed holds significant potential for further research. Modifying the sequence encoding the C-terminal region of the FcRn α-chain would allow generation of cell lines expressing the full-length membrane-bound receptor, opening opportunities for studying intracellular antibody transport and recycling.

Validation of the engineered FcRn using antibody 900 demonstrated that all tested variants bound this antibody with high affinity at pH 6.0 (KD = 3.15–4.13 nM), consistent with the physiological function of FcRn. These results confirm that the engineered receptor retains the ability to recognize IgG with native specificity. Considering the stronger binding of antibody 900 and its high affinity under acidic conditions, it can be hypothesized that this antibody may exhibit an extended in vivo half-life; however, further pharmacokinetic studies are required to verify this hypothesis. Overall, these findings indicate that the recombinant soluble FcRn developed here represents a functional model suitable for in vitro evaluation of IgG–FcRn interactions and can be employed in the development and optimization of therapeutic antibodies with tailored pharmacokinetic properties.

## 5. Conclusions

In summary, this study demonstrates that the developed polycistronic vector pComV-FcRn_B2M, containing nucleotide sequences encoding the P2A peptide in combination with a GSG linker, enables efficient production of functional recombinant FcRn in CHO-K1 cells. For the first time, a strategy for the expression of functional FcRn using polycistronic constructs with 2A peptides has been implemented, ensuring the coordinated co-expression of the α-chain and β_2_-microglobulin from a single transcript. The resulting protein predominantly exists as a monomer in solution and retains the key functional characteristic of the native receptor—pH-dependent binding to IgG. The recombinant soluble FcRn developed in this work thus represents a robust and physiologically relevant model for in vitro screening of Fc-engineered antibodies and for the preliminary evaluation of their pharmacokinetic properties.

## Figures and Tables

**Figure 1 pharmaceutics-17-01463-f001:**
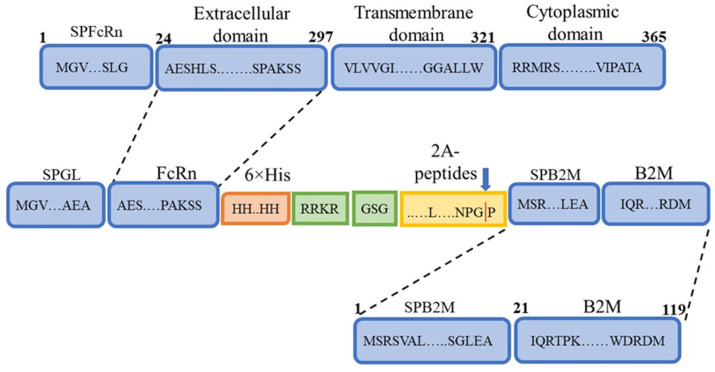
Schematic representation of the designed FcRn heterodimer. SPFcRn—FcRn signal peptide; Extracellular domain—FcRn extracellular domain; Transmembrane domain—FcRn transmembrane domain; Cytoplasmic domain—FcRn cytoplasmic domain; SPGL—Gaussia luciferase signal peptide; 6×His—hexahistidine tag for purification by immobilized metal affinity chromatography (IMAC); RRKR—furin protease cleavage site; 2A peptides—self-cleaving 2A peptides; SPB2M—β2-microglobulin signal peptide; B2M—β2-microglobulin sequence.

**Figure 2 pharmaceutics-17-01463-f002:**
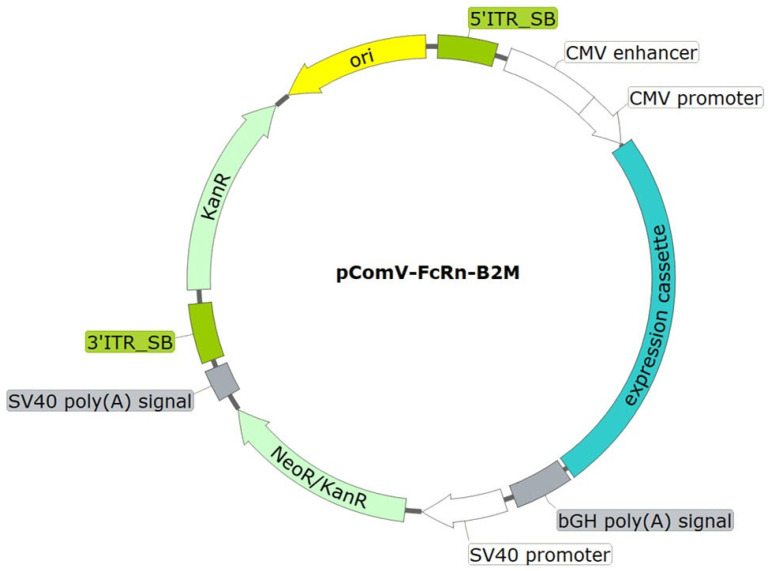
Schematic representation of the integrative plasmid vector pComV-FcRn-B2M. ori—origin of replication; 5′_SB and 3′_SB—SB100X transposase binding sites; CMV promoter—CMV promoter region; bGH poly(A) signal—nucleotide sequence for mRNA stabilization via polyadenylation; NeoR/KanR—nucleotide sequence conferring resistance to kanamycin in bacteria and to neomycin (G418) in eukaryotic cells; SV40 poly(A) signal—nucleotide sequence for mRNA stabilization via polyadenylation.

**Figure 3 pharmaceutics-17-01463-f003:**
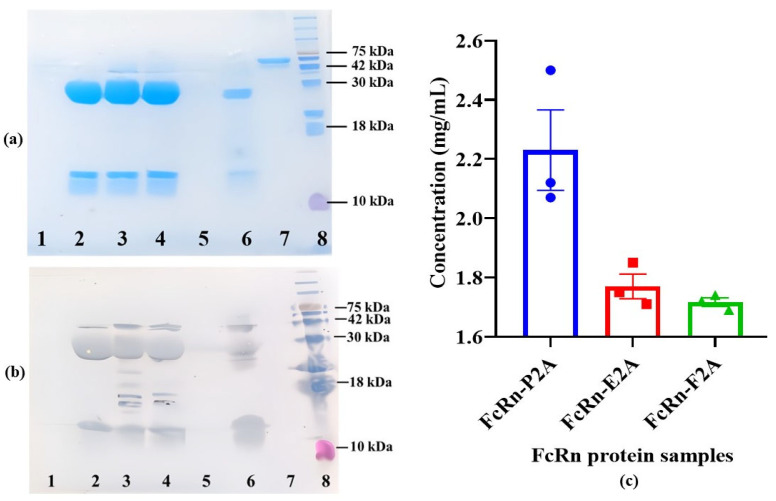
Expression and yield analysis of recombinant FcRn produced using different 2A peptides. (**a**,**b**) SDS–PAGE and Western blot analysis of purified FcRn samples. Lanes: 1—FcRn-P/L; 2—FcRn-P2A; 3—FcRn-E2A; 4—FcRn-F2A; 5—FcRn-T2A; 6—positive control (FcRn); 7—negative control (BSA); 8—molecular weight marker (Prestained Protein Marker II, 10–200 kDa). (**c**) Comparison of recombinant FcRn concentrations in FcRn-P2A, FcRn-E2A, and FcRn-F2A samples. Protein concentrations were determined spectrophotometrically at 280 nm following affinity purification. Data are presented as mean ± standard error of the mean (Mean ± SEM, n = 3).

**Figure 4 pharmaceutics-17-01463-f004:**
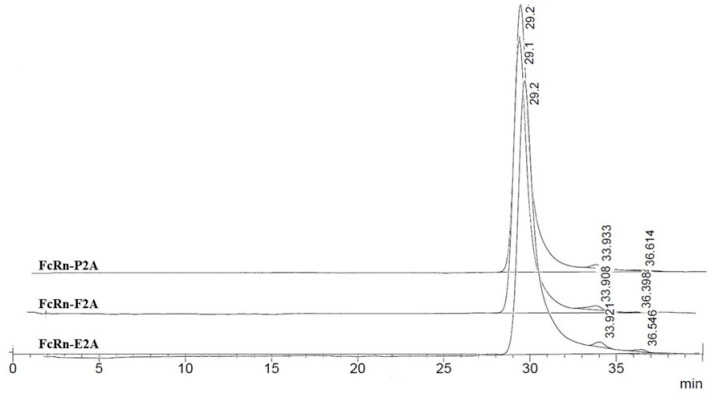
Size-exclusion chromatography (SEC) profiles of FcRn-P2A, FcRn-E2A, and FcRn-F2A samples analyzed on a PROTEIN KW-803 column.

**Figure 5 pharmaceutics-17-01463-f005:**
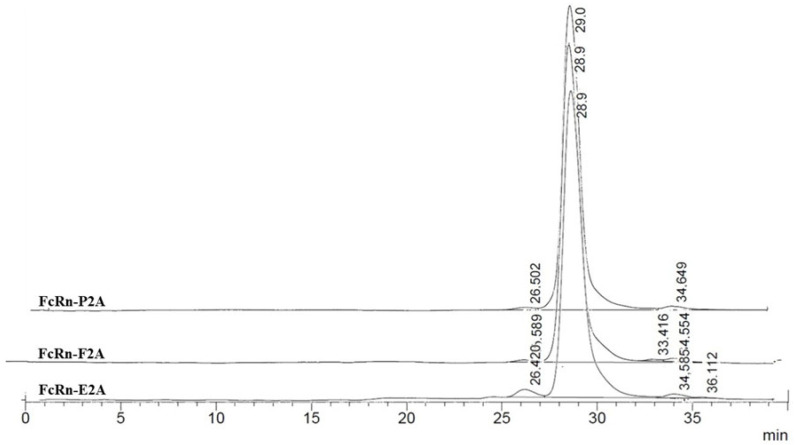
Size-exclusion chromatography (SEC) profiles of FcRn-P2A, FcRn-E2A, and FcRn-F2A samples analyzed on a PROTEIN KW-803 column.

**Figure 6 pharmaceutics-17-01463-f006:**
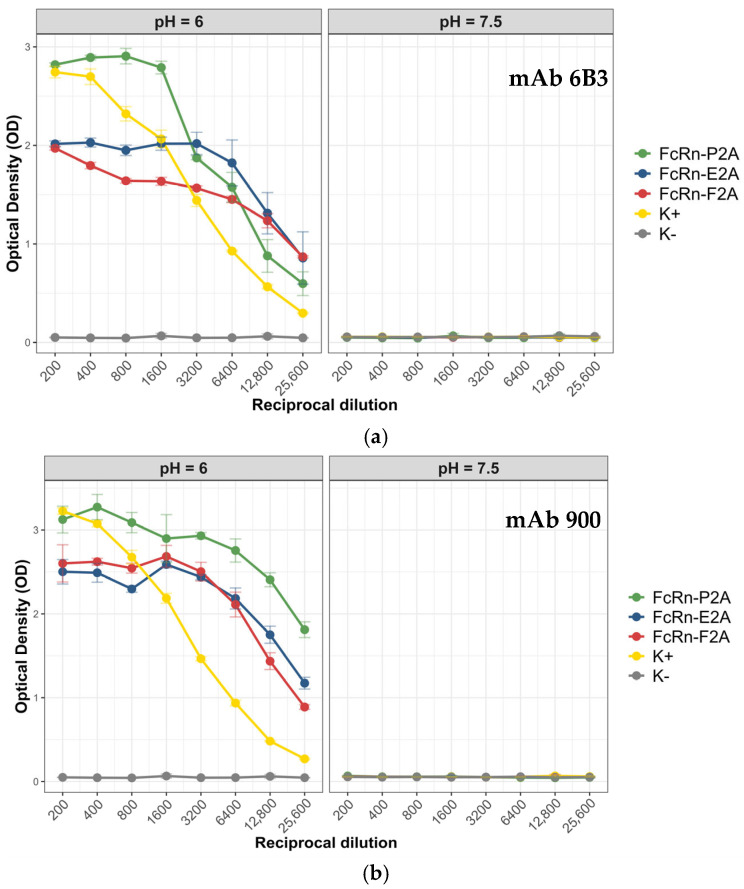
Interaction of recombinant FcRn variants with monoclonal antibodies 6b3 and 900 under different pH conditions. (**a**) Binding of recombinant FcRn variants to antibody 6b3 at pH 6.0 and pH 7.5. (**b**) Binding of recombinant FcRn variants to antibody 900 at pH 6.0 and pH 7.5. Data are presented as mean ± standard deviation (SD, n = 3).

**Table 1 pharmaceutics-17-01463-t001:** Oligonucleotide sequences used for the generation of pComV-FcRn-P2A-B2M, pComV-FcRn-E2A-B2M, pComV-FcRn-F2A-B2M, and pComV-FcRn-T2A-B2M plasmids.

Name	Sequence (5′→3′)
FcRn-F	5′-aaaaaagctagcgaattcgccaccatgggagt-3′
FcRn-R	5′-aaaaaagtcgacttacatgtcgcggtc-3′
FcRn_F2	5′-aaaaagaattcgccaccatgggagtgaaggtgctgttc-3′
FcRn_E2A_R	5′-acctgggttgctctcaacatctccagccaatttcaagagagcataattagtacactgaccagaacccctctttctcctgtgatgatggtggtgatg-3′
B2M_E2A_F	5′-cagtgtactaattatgctctcttgaaattggctggagatgttgagagcaacccaggtcccatgtccagatctgtggccctg-3′
FcRn_P2A_R	5′-agggttctcctccacgtctccagcctgcttcagcaggctgaagttagtggcaccagaacccctctttctcctgtgatgatggtggtgatg-3′
B2M_P2A_F	5′-gccactaacttcagcctgctgaagcaggctggagacgtggaggagaaccctggacctatgtccagatctgtg-3′
B2M_R	5′-aaaagtcgacttacatgtcgcggtcccacttcac-3′
FcRn_F2A_R	5′-ggactcgacgtctcccgccagcttgagaaggtcaaaattcaaagtctgtttcacaccagaacccctctttctcctgtgatgatggtggtg-3′
B2M_F2A_F	5′-cagactttgaattttgaccttctcaagctggcgggagacgtcgagtccaaccccgggcccatgtccagatctgtggcc-3′
FcRn_T2A_R	5′-gccgggattctcctccacgtcaccgcatgttagaagacttcctctgccctcaccagaacccctctttctcctgtgatgatggtggtgatg-3′
B2M_T2A_F	5′-gagggcagaggaagtcttctaacatgcggtgacgtggaggagaatcccggccctatgtccagatctgtg-3′

**Table 2 pharmaceutics-17-01463-t002:** Constructs of polycistronic plasmids encoding FcRn and β_2_-microglobulin linked via different 2A peptides.

Plasmid Name	Protein Designation	Origin of 2A Peptide	Amino Acid Sequence
pComV-FcRn-P2A_LESS-B2M	FcRn-P/L	P2A porcine teschovirus-1	ATNFSLLKQAGDVEENPGP
pComV-FcRn-P2A-B2M	FcRn-P2A	P2A porcine teschovirus-1	GSG*ATNFSLLKQAGDVEENPGP
pComV-FcRn-E2A-B2M	FcRn-E2A	E2A equine rhinitis A virus	GSG*QCTNYALLKLAGDVESNPGP
pComV-FcRn-F2A-B2M	FcRn-F2A	F2A foot-and-mouth disease virus	GSG*VKQTLNFDLLKLAGDVESNPGP
pComV-FcRn-T2A-B2M	FcRn-T2A	T2A thosea asigna virus	GSG*EGRGSLLTCGDVEENPGP

*—linker influencing the mechanism of action of 2A peptides.

**Table 3 pharmaceutics-17-01463-t003:** Binding constants (KD) of recombinant FcRn variants (P2A, E2A, and F2A) with antibody 900 at pH 6.0 and pH 7.5, determined by bio-layer interferometry (BLI).

Protein Designation	pH Value	Antibody 900 Concentration (nM)	KD (nM)	KD Error (nM)
FcRn-P2A	7.5	89.9	17.3	0.36
FcRn-P2A	6.0	89.9	3.15	0.05
FcRn-E2A	7.5	89.9	15.8	0.33
FcRn-E2A	6.0	89.9	3.46	0.04
FcRn-F2A	7.5	89.9	14.3	0.34
FcRn-F2A	6.0	89.9	4.13	0.06

## Data Availability

The original contributions presented in the study are included in the article; further inquiries can be directed to the corresponding author.

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
