# Peer review of "Does a Polycistronic 2A Design Enable Functional FcRn Production for Antibody Pharmacokinetic Studies?"

_pharmaceutics, 2025, doi:10.3390/pharmaceutics17111463_

Round 1

Reviewer 1 Report

Comments and Suggestions for Authors

This manuscript presents an antibody-specific pharmacokinetic study based on an understanding of the oligomeric state of the neonatal Fc receptor. The paper is extensive and provides detailed explanations. However, certain details require further consideration before publication.

1. Introduction (Line 71): Additional explanation is needed regarding the two main strategies.

2. Methods (Table 1): Both uppercase and lowercase letters are used inconsistently. Is this intentional?

3. Section 2.9, Size Exclusion Chromatography: This part should appear earlier in the Methods section, as it belongs to the main sequence of experimental procedures. Placing it at the end is not appropriate.

4. Results (Figure 3b): The blot appears quite smudged. Can this be improved? If not, at least consider minor graphical adjustments to enhance clarity.

5. Figure 6: The standard deviation bars overlap, making it difficult to assess data reliability. Please revise this figure for better readability.

Author Response

Comments 1: Introduction (Line 71): Additional explanation is needed regarding the two main strategies.

Response 1: Thank you for your comment. We have expanded the Introduction to provide a more detailed explanation of the two main strategies for FcRn expression and to emphasize their key differences. The following text has been added to the manuscript:

"Currently, two main strategic approaches are used for the expression of heterodimeric FcRn [23]. The first is based on co-transfection with two independent monocistronic vectors carrying the genes for the α-chain and βâ‚‚-microglobulin under the control of separate promoters. In this case, the synthesis of each subunit occurs independently. The second approach involves the use of bicistronic constructs, in which the expression of each subunit gene is regulated by its own promoter but both genes are included within a single plasmid construct. In this setup, the synthesis of the individual subunits within the cell becomes coordinated."

This addition clarifies the rationale and differences between the two strategies.

Comments 2: Methods (Table 1): Both uppercase and lowercase letters are used inconsistently. Is this intentional?

Response 2: Thank you for pointing this out. The inconsistent use of uppercase and lowercase letters was not intentional; it resulted from copying the sequence names directly from the plasmid maps. We agree that this could be misleading. The notation in Table 1 has now been standardized throughout the manuscript to ensure consistency and clarity.

Comments 3: Section 2.9, Size Exclusion Chromatography: This part should appear earlier in the Methods section, as it belongs to the main sequence of experimental procedures. Placing it at the end is not appropriate.

Response 3: Thank you for this observation. We agree that the description of the Size Exclusion Chromatography procedure should appear earlier in the Methods section. To improve the logical flow, we have moved this subsection to Section 2.7. Size-Exclusion Chromatography (SEC), directly after Section 2.6. Protein Concentration Assessment. The manuscript has been revised accordingly.

Comments 4: Results (Figure 3b): The blot appears quite smudged. Can this be improved? If not, at least consider minor graphical adjustments to enhance clarity.

Response 4: Thank you for your comment. We have applied image enhancement tools to increase the sharpness and contrast of Figure 3b to improve visual clarity. Unfortunately, Western blot images may naturally appear less defined due to the characteristics of the membrane and detection method, and in this case, some smudging cannot be completely eliminated. However, we believe that the main bands of interest are now clearly distinguishable. The revised figure has been included in the updated manuscript.

Comments 5: Figure 6: The standard deviation bars overlap, making it difficult to assess data reliability. Please revise this figure for better readability.

Response 5: Thank you for this helpful observation. We have revised Figure 6 to improve readability. The updated figure has been included in the revised manuscript.

Reviewer 2 Report

Comments and Suggestions for Authors

In manuscript pharmaceutics-3966917, Nesmeyanova et al examine whether polycistronic 2A design enable functional FcRn production for antibody pharmacokinetic studies. The topic of this manuscript is interesting and fits well the scope of pharmaceutics. The reviewer feels it can be accepted after some amendments.

(1) Statistical analysis is missing.

(2) The functional relevance of the recombinant FcRn is only tested with pH-dependent IgG binding. No evidence is provided to confirm whether the receptor is capable of IgG recycling or transport in a cellular context. A cellular bast / in vivo based test may be more appropriate. 

(3) Fig 4/5: The small difference in retention time cannot tell any thing. Even inject same sample, such small variation exists. 

(4) Is such usage appropriate? Acknowledgments: During the preparation of this manuscript, the authors used ChatGPT (GPT-5, 518 OpenAI) for the purpose of translating the Russian version of the article into English. The authors 519 have reviewed and edited the resulting text and take full responsibility for the content of this pub-520 licatio. 

Author Response

Comments 1: Statistical analysis is missing.

Response 1: Thank you for your comment. We have processed the results of the Bio-Layer Interferometry (BLI) analysis. The corresponding revisions have been incorporated into the manuscript.

Comments 2: The functional relevance of the recombinant FcRn is only tested with pH-dependent IgG binding. No evidence is provided to confirm whether the receptor is capable of IgG recycling or transport in a cellular context. A cellular bast / in vivo based test may be more appropriate. 

Response 2: You are correct, and we have addressed this limitation in the Discussion section. The added text reads: “However, a limitation of our study should be considered. We have demonstrated the functionality of the soluble FcRn form in vitro, but it lacks the transmembrane domain and is therefore incapable of replicating the full cellular IgG recycling cycle. This prevents direct modeling of in vivo antibody transport in our system. It is important to emphasize that the polycistronic vector we developed holds significant potential for further research. Modifying the sequence encoding the C-terminal region of the FcRn α-chain would allow generation of cell lines expressing the full-length membrane-bound receptor, opening opportunities for studying intracellular antibody transport and recycling.”

Comments 3: Fig 4/5: The small difference in retention time cannot tell anything. Even inject same sample, such small variation exists.

Response 3: We thank the reviewer for this important remark. We fully agree that small run-to-run shifts in retention time (RT) can occur even for the same injected sample and by themselves are not conclusive. For this reason, our conclusions do not rely on minor RT differences alone. Instead, we interpret the SEC data using three complementary criteria: (i) the position of the chromatographic peak relative to calibrated molecular weight standards (calibration curve: log(MW) vs RT), (ii) the percentage contribution of the main peak to the total chromatogram area, and (iii) the presence of reproducible additional peaks (indicative of aggregates or fragments). In our data, the main peak for all three FcRn variants represents ≥95% of total area and corresponds to the expected monomeric size (~35–40 kDa), while the FcRn-E2A sample reproducibly contains a minor additional peak at ~110 kDa (≈1% area).

Comments 4: Is such usage appropriate?Acknowledgments: During the preparation of this manuscript, the authors used ChatGPT (GPT-5, 518 OpenAI) for the purpose of translating the Russian version of the article into English. The authors 519 have reviewed and edited the resulting text and take full responsibility for the content of this pub-520 licatio.

Response 4: You are correct, and we followed the journal’s template for the Acknowledgments section. The instructions explicitly recommend disclosing the use of GenAI tools as follows: “During the preparation of this manuscript/study, the author(s) used [tool name, version information] for the purposes of [description of use]. The authors have reviewed and edited the output and take full responsibility for the content of this publication.”

Accordingly, we included: “During the preparation of this manuscript, the authors used ChatGPT (GPT-5, OpenAI) for the purpose of translating the Russian version of the article into English. The authors have reviewed and edited the resulting text and take full responsibility for the content of this publication.”

Round 2

Reviewer 1 Report

Comments and Suggestions for Authors

The authors have responded sincerely to the reviewers’ comments.
The manuscript is now ready for publication.